# A Cruciform Petal-like (ZIF-8) with Bactericidal Activity against Foodborne Gram-Positive Bacteria for Antibacterial Food Packaging

**DOI:** 10.3390/ijms23147510

**Published:** 2022-07-06

**Authors:** Bowen Shen, Yuxian Wang, Xinlong Wang, Fatima Ezzahra Amal, Liying Zhu, Ling Jiang

**Affiliations:** 1State Key Laboratory of Materials-Oriented Chemical Engineering, College of Biotechnology and Pharmaceutical Engineering, Nanjing 211816, China; 202162118004@njtech.edu.cn (B.S.); yxwang@njtech.edu.cn (Y.W.); 2College of Food Science and Light Industry, Nanjing Tech University, Nanjing 210009, China; wangxinlong@njtech.edu.cn (X.W.); wazir@mailnino.com (F.E.A.); 3School of Chemistry and Molecular Engineering, Nanjing Tech University, Nanjing 210009, China

**Keywords:** cruciform petal-like ZIF-8, foodborne bacteria, antibacterial, ZIF-8-Film, food preservation, food packaging

## Abstract

Most antibacterial nanomaterials used in food packaging act by releasing reactive oxygen species (ROS), which cannot efficiently have an inhibitory effect by penetrating the cell wall of Gram-positive *Staphylococcus aureus*. In this work, we used the cruciform petal-like zeolite imidazole framework-8 (ZIF-8) synthesized in the water phase which can release active Zn compounds in aqueous solution and exert a stronger inhibitory effect on *S. aureus*. The experimental results demonstrated that the aqueous cruciform petal-like ZIF-8 has the same photocatalytic activity as traditional ZIF-8 and can be applied in photocatalytic bacterial inactivation. The cruciform petal-like ZIF-8 was also shown to release active Zn compounds in aqueous solution with a better antibacterial effect against *S. aureus,* reaching 95% inactivation efficiency. The antibacterial effect was therefore 70% higher than that of traditional ZIF-8. Based on its excellent antibacterial properties, we loaded petal-like ZIF-8, PDA and PVA onto ordinary fibers to prepare ZIF-8-Film. The results further showed that ZIF-8-Film has a high filtration capacity, which can be used in antibacterial packaging material with the required air permeability. Moreover, ZIF-8-Flim can clean the surface on its own and can maintain a sterile environment. It is different from other disposable materials on the market in that it can be reused and has a self-disinfection function.

## 1. Introduction

The World Health Organization showed that food safety has become a serious public health problem all over the world [1,2] and foodborne bacteria contaminating food is a common food safety issue [3,4]. It appears that preventing foodborne bacteria has become even more difficult since the overuse of antibiotics and the subsequent rise of drug resistance especially that represented by Gram-positive *Staphylococcus aureus* [5,6]. Therefore, nanomaterials have gradually become new antibacterial materials and antibacterial films based on nanomaterials are gradually being used in food packaging [7]. Antibacterial food packaging systems can inhibit the growth of bacteria to preserve the safety and quality of foods during transportation and storage, which represents an innovative and promising food packaging technology [7,8].

The rapid development of nanomaterials technology has recently become a new means of strengthening the antibacterial ability of food packaging [7,9]. Although the Au NCs previously studied by our team have excellent antibacterial properties, their inhibitory effect against gram-positive bacteria is limited [10]. According to the literature, this phenomenon is common, with reports indicating that Gram-positive bacteria can withstand ROS by nanomaterials due to the features of their cell wall and cell membranes [11,12,13,14]. ZIF-8 is a MOF that was created by combining zinc as a metal coordination center with 2-methylimidazole as a ligand [15], which has excellent properties and can be applied in a variety of research fields [16,17]. Traditional ZIF-8 has potent photo-induced bactericidal activity, which can effectively kill *Escherichia coli* (*E. coli*) due to ROS generation in the presence of light [18]. Furthermore, traditional ZIF-8 can also release zinc ions and 2-methylimidazole to kill *E. coli* effectively under a variety of environmental conditions [19]. However, light-driven inactivation of bacteria as a photocatalyst is the main source of antibacterial ability in traditional ZIF-8 [20,21,22]. However, ZIF-8 has not been directly used to inhibit Gram-positive *S. aureus* to date. Fang et al. [23] discovered that small Pd NPs are more likely to penetrate *S. aureus* cells and cause more damage to the bacteria. Therefore, whether dissociation of ZIF-8 in a solution can inhibit *S. aureus* is crucial for assessing its antibacterial potential.

ZIF-8 synthesized in the water phase is easier to dissociate in aqueous solution [24,25]. For this study, we synthesized petal-like ZIF-8 (P-ZIF-8) in the water phase. As a control, we chose the typical dodecahedral ZIF-8 (D-ZIF-8) synthesized in the methanol phase [26]. We first validated the photocatalytic ROS generation ability of P-ZIF-8, followed by an assessment of its dissociation in aqueous solution, and finally found the inhibitory effect of free active Zn compounds for Gram-positive bacteria different from ordinary zinc ions. We offer a possible mechanistic explanation for these experimental results based on a combination of experimental and theoretical techniques. Finally, ZIF-8 was integrated into a fiber film which exhibited outstanding performance in complete food safety management and personal health protection due to its antibacterial characteristics. This work offers a reference for further research on the inhibition of foodborne Gram-positive bacteria, clarifies a new idea and method for food safety packaging and contributes to the development and application of food packaging.

## 2. Results and Discussion

### 2.1. Characterization of ZIF-8

ZIF-8 crystallites of different forms were prepared and used for structure tests. The images of the two forms of ZIF-8 are shown in Figure 1A,B. The ZIF-8 synthesized in methanol had a dodecahedral structure with a crystal size of about 200–300 nm, while the ZIF-8 synthesized in the water phase had a petal shape with a crystal size of about 3–5 μm. D-ZIF-8 had a uniform shape with little variation between individual particles while P-ZIF-8 exhibited significant variation and a more cluttered appearance, indicating that it may more easily dissociate in aqueous solution (Appendix A). By contrast, ZIF-8 with its dodecahedral shape had a more uniform morphology and higher crystallinity. The solvent can impact the growth of ZIF-8 particles, resulting in various crystal morphologies, even when the reactant ratio is the same [27]. The corresponding XRD patterns shown in Figure 1C, confirmed the difference between the two forms of ZIF-8. It can be seen that there are regular peaks in the XRD diffraction pattern of D-ZIF-8 whereby diffraction peaks of (110), (200), (220), (310) and (222) can be seen at 7–20°, which is also consistent with the regular crystal plane of D-ZIF-8 [28]. The P-ZIF-8 had a crystalline structure as well, but the diffraction peaks were very disorderly, indicating that the surface had an uneven distribution, which was consistent with the observed surface morphology [29]. The FTIR data (Figure 1D) revealed that the two forms of ZIF-8 had the same functional groups. The infrared peaks of the two versions of ZIF-8 were almost identical. Near 1600 cm^−1^, the C=N tensile mode was detected, while the double peak at 1400–1500 cm^−1^ was attributed to the bending vibration of -CH_3_, and the peak at 1100–1200 cm^−1^ is categorized as C-N. The C-H bending vibration has been linked to in-plane stretching of about 1000 cm^−1^ and 500–1000 cm^−1^ [30]. ZIF-8 in both forms had a high light absorption capacity [31], as observed using an ultraviolet spectrophotometer (Appendix A). ZIF-8 exhibited a stronger sensitivity to visible light irradiation, which is related to its photocatalytic ability [32]. Therefore, we also investigated the capacity of two forms of ZIF-8 to photodegrade methylene blue (MB) in the presence of light. As expected, MB was degraded by ZIF-8, resulting in a reduction of the measure absorbance with time. Both forms of ZIF-8 degraded methylene blue within a short time, as seen in Figure 1E. This result shows that ZIF-8 can better absorb light and convert photons into active molecules that induce the decomposition of methylene blue. P-ZIF-8 had a good photocatalytic capacity despite its structure and shape differing from D-ZIF-8.

### 2.2. Antibacterial Activity against Foodborne Bacteria of ZIF-8

After initial characterization and simple experiments, the light absorption ability of P-ZIF-8 was confirmed, which prompted us to investigate its antibacterial activity. The Gram-negative bacterium, *E. coli* and the Gram-positive bacterium, *S. aureus,* were both inhibited by both forms of ZIF-8. Although both materials had the ability to kill the tested bacteria at increasing concentrations, the concentrations at which they can inhibit bacterial growth differed. The MIC (0.5 mg/mL) and MBC (1 mg/mL) of D-ZIF-8 and P-ZIF-8 against *E. coli* were almost identical, showing that both materials have a good inhibitory effect on *E. coli*. However, the results revealed that *S. aureus* had different sensitivity to D-ZIF-8 and P-ZIF-8, indicating that the same inhibitory effect did not apply to all bacteria. P-ZIF-8 had a lower MIC (0.5 mg/mL) and MBC (1 mg/mL) against *S. aureus* than D-ZIF-8 (MIC = 1 mg/mL, MBC = 2 mg/mL), indicating that P-ZIF-8 is more effective in inhibiting *S. aureus* (Appendix A and Appendix A).

Furthermore, the instant antibacterial effect on bacteria treated with different concentrations of materials was used to evaluate the antibacterial activity of the two forms of ZIF-8. The overall viability of the bacteria proved the concentration-dependent antibacterial activity of ZIF-8 (Figure 2A–C). We clearly noticed that the killing efficiency increased as the concentration of ZIF-8 increased. The overall inhibitory effect of D-ZIF-8 on bacteria was basically the same. With the increase in D-ZIF-8 concentration, the Gram-negative *E. coli* and Gram-positive *S. aureus* showed the same proportion of cell death, but with the increase in P-ZIF-8 concentration, more Gram-positive *S. aureus* cells were inactivated This result confirmed that P-ZIF-8 has a distinct effect against the Gram-positive *S. aureus*. According to previous studies, some MOFs are more effective against Gram-positive than against Gram-negative bacteria [11,33]. The authors hypothesized that the outer membrane of Gram-negative bacteria can prevent MOFs from diffusing into their cells [34]. After antibacterial treatment with MOFs, SEM pictures revealed many pits and nicks in the cell membranes of Gram-positive bacteria, but no significant alteration in the cell membranes of Gram-negative bacteria [33]. Overall, MOF particles have the ability to enter cells and cause damage and Gram-positive bacteria are vulnerable to this invasion due to the lack of an outer cell membrane [33,34]. Through antibacterial experiments, it is believed that P-ZIF-8 can more effectively fight foodborne Gram-positive bacteria.

### 2.3. Antibacterial Mechanism of ROS Generation by ZIF-8

To investigate the antibacterial mechanism of ZIF-8, we first confirmed its ability to generate ROS, followed by an analysis of other possible antibacterial mechanisms. Many scientists believe that the antibacterial activity of ZIF-8 stems from its photocatalytic capacity [18,30,35]. The Ligand to metal charge transfer (LMCT) process generates photoelectrons, which are collected as paramagnetic Zn^+^ sites on the surface of ZIF-8 and subsequently transported from the Zn^+^ center to O_2_ and H_2_O [18,36]. This process generates ROS, which aids the photocatalytic inactivation of microorganisms [20]. Since H_2_O_2_ is thought to be the principal active antibacterial ROS, we tested the capacity of ZIF-8 to generate H_2_O_2_ in the presence of light using 3,3,5,5-tetramethylbenzidine (TMB) [37]. Peroxidases accelerate the breakdown of H_2_O_2_, while also oxidizing TMB to TMB^+^, which exhibits an absorption peak at 652 nm [18]. According to the UV-Vis spectroscopy measurements, the photocatalytic H_2_O_2_ production of ZIF-8 increased with time (Figure 3A,B). The results indicated that both forms of ZIF-8 can create H_2_O_2_ quickly when exposed to light and swiftly convert TMB blue when activated by peroxidase. P-ZIF-8 is a quick catalyst that generates H_2_O_2_ under illumination. The presence of superoxide anions(•O_2_^−^) can be detected using nitrotetrazolium blue chloride (NBT). The •O_2_^−^ anions can combine with NBT in DMSO solution to produce a blue-violet compound that can be detected at 529nm [38]. We discovered the presence of •O_2_^−^ in solutions of both forms of ZIF-8 when they are illuminated **(**Figure 3C,D). The fluorescent probe TA was next used to measure the level of •OH created by ZIF-8 in aqueous solution. When hydroxyl radicals are present in the solution, TA is oxidized to TAOH, which exhibits a fluorescence signal at 420 nm [23]. The two types of ZIF-8 both oxidized TA in a short amount of time (Appendix A). Because ZIF-8 cannot form •OH, the breakdown of the generated H_2_O_2_ in the solution is the source of •OH [18]. D-ZIF-8 exhibited a small advantage in the ability to produce ROS, which corresponds to its small advantage in inhibiting *E. coli*. We evaluated the quantity of ROS in cells to see if bacterial cell death is caused by oxidative damage, in order to see if the ROS produced by ZIF-8 plays a crucial role in suppressing bacteria. Bacteria and positive reagents were used in the positive control group, bacteria and water were used in the negative control group and bacteria in various types of material solutions were used in the experimental group. The measurement results are shown in Figure 3E,F, the negative control group of *E. coli* had low fluorescence intensity, indicating that the ROS concentration in untreated cells was low. The ROS levels of the positive control group were 3-fold higher than in the negative control group. The quantity of ROS in the bacterial cells rose quickly after treatment with the positive control reagent, demonstrating that ROS can readily permeate the cells, inducing internal oxidative damage and cell death. The data further demonstrates that the ZIF-8-treated bacteria had more ROS inside the cells. Furthermore, the amount of intracellular ROS produced by D-ZIF-8 and P-ZIF-8 exhibited a slight difference and D-ZIF-8 apparently had a stronger ability to cause oxidative damage to bacteria. Nevertheless, P-ZIF-8 also had an appreciable ability to cause oxidative damage to bacteria. It is worth noting that the amount of ROS in *S. aureus* cells was lower than in *E. coli*, indicating that the cell structure of *S. aureus* offers better protection against ROS penetration, as previously reported [10]. Both D-ZIF-8and P-ZIF-8 were, therefore, able to absorb light to form active compounds, act on different bacteria and induce cell death, but the unique antibacterial mechanism of P-ZIF-8 required further analysis.

### 2.4. Antibacterial Mechanism of ZIF-8 Involves Active Zn Compounds

We demonstrated that when ZIF-8 is exposed to light, it creates antibacterial active chemicals. P-ZIF-8 can also produce ROS for antibacterial purposes, but this does not explain why it is able to inhibit *S. aureus* more effectively. To further investigate the antibacterial mechanism of P-ZIF-8, we studied bacteria morphology and deduced the effect of the materials on bacteria cells. Figure 4A allows us to see the morphology of bacteria in great detail. The native morphology of *E. coli* is a smooth cylindrical form that is found in the undamaged cell. The surface of *E. coli* is wrinkled and some cells are destroyed following ZIF-8 treatment, suggesting that the cell surface is substantially impacted. The *E. coli* treated with P-ZIF-8 exhibited cell morphological damage and no P-ZIF-8 original form was found among the *E. coli* cells, however, some morphologically diverse substances were visible. By contrast, *E. coli* exposed to D-ZIF-8 exhibited more damage, with some full D-ZIF-8 particles remaining in the field of vision. Intact *S. aureus* cells are smooth and spherical and their morphology can be well preserved. After exposure to the ZIF-8 solution, there was little change, indicating that *S. aureus* cells are more resistant to the treatment. A small number of broken cells was found in the field of vision, with more broken cells among the bacteria exposed to P-ZIF-8. The microscopy indicated that D-ZIF-8 particles remain intact, while P-ZIF-8 appears to be degraded during treatment. ZIF-8 is an MOF with a metal core of Zn^2+^. According to prior research, Zn^2+^ has antibacterial properties and can attach to active proteins on the bacterial cell surface and inside the cells, leading to inactivation [39]. We examined the capacity of different forms of ZIF-8 to release Zn^2+^ in aqueous solutions in order to better understand the Zn^2+^ mechanism. According to research by Taheri et al. [19], ZIF-8 has varying Zn^2+^ release capacities in different solvents, which is one of the major determinants in bacterial inactivation. We speculated that the enhanced antibacterial activity of P-ZIF-8 may be due to the release of active Zn compounds [19]. To test this idea, we measured the free zinc concentration of the two forms of ZIF-8 in aqueous solution. The zinc concentration in the aqueous solution was measured by ICP after 1 h of incubation. The measurement results revealed that ZIF-8 releases free zinc in aqueous solution. Nevertheless, the aqueous solution of P-ZIF-8 clearly had clealy more free zinc (Figure 4B). We are unsure if ZIF-8 will dissociate in aqueous solution into a large number of smaller particles. Although there have been indications that ZIF-8 can dissociate in various solutions to release zinc, it is unclear in which chemical form [40]. The original form of P-ZIF-8 can be completely preserved by storing it in an ethanol solution. When P-ZIF-8 was agitated for 1 h in an aqueous solution, it lost its original form and appeared as many small particles, which may be able to inflict more harm to *S. aureus* (Figure 4C). Further thermogravimetric analysis (Figure 4D) indicated that P-ZIF-8 is more unstable and prone to disintegrate, which is consistent with its dissociation into active Zn compounds in aqueous solution. D-ZIF-8 lost about 2% of water at 25–150 °C, while P-ZIF-8 lost about 15% of crystal water at 25–150 °C. The result at a high temperature of 800 °C was that D-ZIF-8 had 60% of its weight remaining, while P-ZIF-8 had less than 50% remaining. The DTG results showed that P-ZIF-8 had two distinct thermal decomposition activities, while D-ZIF-8 had only one, which indicated that P-ZIF-8 had worse thermal stability (Appendix A). The antibacterial properties of the aqueous solution of Zn(NO_3_)_2_ employed in the production of ZIF-8 were also investigated (Figure 4E). The antibacterial properties of Zn^2+^ have been extensively reported and our assay confirmed that Zn^2+^ has an impact on both Gram-negative and Gram-positive bacteria. However, the inhibitory effect of Zn^2+^ on Gram-positive and Gram-negative bacteria was similar. Zn^2+^ binds to proteins on or within the cell membrane, causing a disruption of the normal physiological activities of the bacteria [41]. Both Gram-negative and Gram-positive bacteria may absorb Zn^2+^ without discrimination and pure Zn^2+^ has no preferential inhibitory effect on Gram-positive bacteria [42].

ZIF-8 has also been shown to release Zn_x_L_y_ (L = 2-methyl imidazole) compounds in aqueous solution [43], which may have a role in its enhanced antibacterial activity [40]. We filtered the larger P-ZIF-8 nanoparticles, leaving only the supernatant as a bacteriostatic agent, whereby the small particles of active Zn compounds stay in the supernatant. The results indicate that the extract of P-ZIF-8 had a stronger inhibitory effect on *S. aureus* (Figure 5A), suggesting that the free zinc residue of P-ZIF-8 in the solution plays a key antibacterial role. In order to detect the free zinc that has entered the cells, we mixed the P-ZIF-8 supernatant with *S. aureus* for 1 h and then lysed the cells, after which the Zincon monosodium salt [2-(2-Hydroxy-5-sulfophenylazo) benzylidene hydrazinobenzoic acid] was used to evaluate the zinc concentration at pH = 8. The Zincon monosodium salt and zinc can interact to generate a blue complex that can be quantified at 620 nm [44]. As shown in Figure 5B, the *S. aureus* cells treated with P-ZIF-8 took up considerably more zinc than those treated with D-ZIF-8. According to Fang et al. [23], tiny nanoparticles may more easily enter the cell interior and cause damage to *S. aureus* cells due to differences in cell membrane permeability, resulting in a higher inhibitory impact of such nanomaterials on *S. aureus*. This might explain why P-ZIF-8 was more effective against *S. aureus*. Our results, therefore, indicate that ZIF-8 can produce ROS while also releasing active Zn compounds. However, P-ZIF-8 causes greater damage to *S. aureus* due to its greater propensity to dissociate in aqueous solution. Thus, P-ZIF-8 has unique antibacterial properties that can be applied further in the field of food packaging.

### 2.5. Synthesis and Filtering Ability of ZIF-8-Film

A simple synthesis technique was used to create the ZIF-8-Film using P-ZIF-8, polydopamine (PDA) and polyvinyl alcohol (PVA) (Figure 6A). As demonstrated in the SEM image (Figure 6B–E), ZIF-8 nanoparticles were effectively loaded onto the surface of the fiber. PDA applied throughout the preparation process can increase the antibacterial properties of the fabric and is simple to functionalize, making it ideal for loading nanomaterials onto textile fibers. The use of polyvinyl alcohol (PVA) can increase the viscosity of the mixture, allowing it to adhere securely to the surface of the fiber [45]. Bacteria and viruses generally cannot be airborne on their own and are instead transported inside very small aqueous respiratory droplets [46]. Therefore, viruses and bacteria are usually transmitted to food through small water droplets [47]. We tested the water contact angle of a typical fiber. As indicated in the diagram (Appendix A), the water contact angle of a 20 μL drop on a typical fiber is approximately 80°. This result demonstrates that the surface of a typical fiber is smooth, hydrophobic and devoid of nanostructures. Despite its low hydrophobicity, bacteria-laden droplets may nevertheless stick to the original fiber’s surface. After manufacturing the ZIF-8-Film, ZIF-8 nanoparticles are loaded onto the fiber surface, improving the hydrophobic characteristics of fiber, reducing droplet adsorption and making them suitable for food packaging. The contact angle of the ZIF-8-Film material was 105° (Appendix A), indicating a substantial improvement in hydrophobicity. It is possible that the improved hydrophobicity is related to the surface charge collected at the nanomaterial’s edge, which increases the repulsive force between fibers and water droplets [46]. Improved hydrophobicity efficiently prevents droplets containing poisons (including viruses and germs) from sticking to the fiber, enhancing protection and reducing water loss inside food packing [48]. To test the filtration capacity of the treated fabric, we created a filter test model, as illustrated in Appendix A. The left space was sprayed with a bacteria-containing spray (10^5^ CFU/mL) for 10 min at a speed of 0.5 m/s, while the right space was a sterile and dry environment. After 12 h, no colonies formed on agar plates in the right region (Appendix A). ZIF-8-Film was effective against both *E. coli* and *S. aureus*, demonstrating that it has a high filtering ability and can eliminate germs. At the same time, the mechanical filtering capacity of the ZIF-8-Film was also tested. PM_2.5_ particles were introduced into the left chamber and the same was isolated with a ZIF-8-Film in the middle. In the right chamber, a collector was used to measure the residual PM_2.5_. When the flow rate was set to 0.5 m/s, the removal rate of PM_2.5_ by the ZIF-8-Film reached 98% and when the flow rate was increased to 1.0 m/s it remained at 92% (Figure 7A), which shows that the filtration ability of the ZIF-8-Film is very good and comparable to its ability to remove bacteria. ZIF-8 Film has excellent filtration and waterproof ability, which is usually necessary for food packaging, and which means ZIF-8 film has unique advantages in food packaging.

### 2.6. Antibacterial Activity of the ZIF-8-Film

The ZIF-8-Film should not only be able to filter germs, but it should also be highly antimicrobial. As a result, we focused on the antibacterial properties and long-term use performance of the ZIF-8-Film. As previously reported, the combination of nanomaterials and polymers tends to have better antibacterial capabilities [49,50]. The ZIF-8-Film was cut into round pieces and immersed in suspensions containing different concentrations of bacterial cells, followed by transfer to agar plates for colony counting. The incubated ZIF-8-Film was then subjected to repeated antibacterial assays to verify its long-term use performance. The inhibition rate of the ZIF-8-Film remained at a high level in the first five cycles, as shown in Figure 7B,C, suggesting that our ZIF-8-Film can be reused and retain strong antibacterial performance with repeated usage. This allows ZIF-8-Film to be used for food preservation since it can remove germs on its own and preserve the sterile environment within the film. We hypothesized that there were bacteria attached to the film surface and therefore tested whether ZIF-8-film could kill the bacteria attached to its surface. We dripped the bacterial suspension on the dry filter and incubated it in dry air for 3 h before transferring it to LB liquid medium for cultivation. The bacteria grew strongly after the raw fibers were transferred to the medium and cultured for 12 h. By contrast, the ZIF-8-Film that was transferred to the culture medium did not contain any viable bacteria (Appendix A), indicating that it has a very strong self-cleaning function and can neutralize bacteria on its own. We also placed the filter solid medium that had been inoculated with bacteria to see if it could eliminate the bacteria that had been introduced. Colonies proliferated on the solid medium covered by the raw fibers after 12 h, whereas no colonies developed on the solid medium covered with the ZIF-8-Film (Appendix A). This result further demonstrated that ZIF-8-Film has strong bactericidal activity. Usually, the antibacterial activity of food packaging can keep food fresh and safe from bacteria. However, ZIF-8-Film may also come into direct contact with food or be absorbed orally. In vitro cell culture assays were used to assess the safety of ZIF-8-Film (Figure 7D). The cell survival rate was reduced when different concentrations of ZIF-8/PDA/PVA mixed solution were added, but it remained above 80%. This means that our ZIF-8 Film is also safe for the human body and there is no need to consider the harm to the human body caused by the residue of food packaging. All in all, our ZIF-8 Film can remove harmful bacteria, maintain the freshness and safety of food and, more importantly, ZIF-8 Film shows the characteristics of being harmless to the human body, eliminating people’s concerns about the safety of food packaging. The strong filtering ability of ZIF-8-Film, as well as its reusable antibacterial capacity, indicate that it has potential food preservation applications in the future.

## 3. Materials and Methods

### 3.1. Strains and Reagents

The reagents were purchased from commercial sources and can be used without further purification. *Escherichia coli* (*E. coli* ATCC 25922) and *Staphylococcus aureus* (*S. aureus* ATCC 25923) were obtained from the Guangdong Microbial Culture Collection Center, Guangzhou, China.

### 3.2. Synthesis of ZIF-8

The P-ZIF-8 was synthesized using the same technique as previously described [51]. The aqueous solutions of Zn(NO_3_)_2_ and 2-methylimidazole were mixed in a 1:10 ratio and stirred for 12 h, after which the white solid was collected by centrifugation, washed three times with ethanol and dried at 80 °C. The aqueous solution in the previous step was replaced with a methanol solution to prepare the D-ZIF-8.

### 3.3. Materials Characterization

The as-prepared powder and the morphology of bacteria exposed to the powder were imaged using a scanning electron microscope (SEM, S4800) with a 20 kV acceleration voltage. The material sample was dispersed in ethanol, dried on a silicon plate, sprayed with gold to increase conductivity, scanned and photographed. The bacterial samples were fixed for 12 h in 2.5% glutaraldehyde solutions, centrifuged, washed for 30 min in a gradient of ethanol solutions and then dripped onto a silicon plate for scanning and imaging. For X-ray diffraction (XRD) analysis, the dried product was placed on a quartz glass slide (Rigaku Miniflex 600). The scanning range was 5–60 degrees, with a step size of 0.008 degrees and a scanning speed of 3 degrees per minute. For transmission electron microscopy (TEM), an ultrasonic bath was used to disperse the crystals in the ethanol solution, followed by a homogenous diffuser to thoroughly disperse the material in the ethanol. The copper net was then placed in an ethanol bath and the analyte examined under a microscope. For Fourier transform infrared spectroscopy the ZIF-8 powder was mixed with the dry KBr powder in an agate mortar, ground finely and placed under a heating source to dry off any remaining water. The mixed powder was then pressed into a pellet and scanned using the instrument. The amounts of zinc cations were calculated using a multi-element standard external calibration curve and the concentrations of released Zn^2+^ ions in various media were measured using an inductively coupled plasma optical emission spectrometry (ICP-OES) instrument (5110, Agilent, Santa Clara, CA, USA). TGA was performed in air with an alumina crucible on a Netzsch STA 449 F3 thermogravimetric analyzer; the mass of the sample was 30 mg and the heating rate was 10 °C/min.

### 3.4. Antibacterial Assays

#### 3.4.1. Cultivation of Bacteria

LB medium (10 g NaCl, 10 g tryptone and 5 g of yeast powder in 1 L of deionized water) was used to grow the bacteria for 12 h until they reached a stable stage, corresponding to 10^9^ CFU/mL.

#### 3.4.2. Instant Antibacterial Effect

The bacteria cultures were diluted to 10^7^ CFU/mL and material solutions of various concentrations were added, followed by incubation for 1 h. After that, the bacteria were diluted to 10^3^ CFU/mL and 100 μL of was spread on LB agar plates, after which the survival rate was determined by counting the colonies. All antibacterial assays were carried out in the presence of visible light.

#### 3.4.3. Measuring the Minimum Inhibitory Concentration (MIC) and Minimal Bactericidal Concentration (MBC)

For this assay,100 μL of material solutions of various concentrations were added to a 96-well plate, followed by a bacterial solution diluted to 2 × 10^6^ CFU/mL with LB medium. To record the growth curve. a microplate reader was used to measure the OD of the bacteria in the 96-well plate at 600 nm (OD_600_) every 2 h. The pure material solution was used to measure the OD_600_ value in a 96-well plate for 12 h and the measurement result of the bacterial solution and material were used to directly deduct the OD_600_ value of the pure material. The MIC was defined as the lowest concentration at which the growth curve does not show a growing trend. After 12 h of culture, the bacterial suspensions with no growth were spread on a solid medium, and the lowest concentration that had no colonies on a solid medium was recorded as the MBC.

### 3.5. Photocatalysis Experiment

As a photocatalytic substrate, a 5 mg/L methylene blue aqueous solution was produced. The photocatalysis experiment was carried out by pouring 30 mL of methylene blue into a quartz photocatalytic tube, adding 30 mg of ZIF-8 as the photocatalyst and reacting for 3 h under the photocatalysis apparatus. The absorbance at 652 nm of the methylene blue aqueous solution was measured at various periods. Methylene blue exhibits a unique UV absorption peak at 652 nm. The Lambert–Beer law states that the change in absorbance may be used to determine the change in methylene blue in the photocatalytic process.

### 3.6. Determination of H_2_O_2_ Produced by ZIF-8

To detect the presence of H_2_O_2_, the H_2_O_2_ produced by ZIF-8 under light was reacted with peroxidase. Ten μL of TMB (80 mM) and 10 μL of HRP (50 μg/mL) were added into the 1 mg/mL of ZIF-8 aqueous solution under the light.The absorption wavelength at 652 nm was then measured and the absorbance was positively related to H_2_O_2_ production.

### 3.7. Superoxide Anion Detection

The NBT (100 μg/mL) of DMSO solution (1 mL) was added to 10 mL (1 mg/mL) ZIF-8 aqueous solution. The superoxide anion produced by ZIF-8 under light will make the originally pale yellow NBT purple and its color depth was positively correlated with the solubility of superoxide anions. The absorbance at 529 nm was measured under an ultraviolet spectrophotometer every 10 min to qualitatively analyze the presence of superoxide anions.

### 3.8. Determination of Hydroxyl Radical

Hydroxyl radicals may react with terephthalic acid (TA) in aqueous solution, transforming it into 2-hydroxy TA (TAOH), which has a bright fluorescence. The emission fluorescence wavelength intensities were measured at different times after dissolving 30 mg of TA in 100 mL of water and adding 30 mg of material to 100 mL of TA solution. The excitation wavelength was set to 315 nm and the intensity of the emission fluorescence wavelength between 350 nm and 600 nm in a fluorescence spectrophotometer was measured.

### 3.9. Intracellular ROS Detection

The ROS content in the cells was measured using the DCFDA detection kit. The bacteria solution (1 mL) was diluted to 10^8^ CFU/mL and a 10 mM DMSO solution of DCFDA (10 μL) was added. ZIF-8 was then added. After incubation under light for 3 h, it was excited at 315 nm and the emission wavelength was detected at 350 nm.

### 3.10. Antibacterial Assays of Zine Ion

In order to avoid additional experimental errors, zinc nitrate was selected as the source of zinc ions. We diluted the bacterial solution (1 mL) to 10^7^ CFU/mL, then added 300, 500 and 700 μL of 1M Zn(NO_3_)_2_ aqueous solution, incubated it for 1 h and then inoculated it on the plate and counted colonies.

### 3.11. Antibacterial Assays of MOF Extract

As in the above experimental process, we only needed to add MOF extract to the bacterial solution. The MOF extract was obtained by dispersing 10 mg/mL ZIF-8 in aqueous solution, then incubating for 1 h and filtering out large particles to obtain a supernatant.

### 3.12. Preparation of ZIF-8-Film

The fiber was soaked in a polydopamine aqueous (1 mg/mL) solution for 12 h, then the washed with ethanol to remove the remaining polydopamine and finally, the fiber was allowed to dry naturally. To obtain ZIF-8-Film, the fiber was soaked in a PVA/ZIF-8 mixed solution for 12 h and then allowed to dry naturally. The mixed solution was made by dissolving 1 g of PVA in 50 mL of water and heating it at 80 °C for 1 h, then 50 mg of ZIF-8 was mixed in 50 mL of water. To make a PVA/ZIF-8 mixed solution, the aqueous solution and the ZIF-8 solution were combined in a 1:1 ratio). PVA connected ZIF-8 to the surface of the fiber and also formed a thick coating on the fiber surface.

### 3.13. MTT Assay

MTT assay was performed to evaluate the ZIF-8/PVA/PDA. In detail, 3T3 cells were seeded in 96-well plates with a density of 104 cells per well and incubated in Dulbecco’s modified Eagle medium (DMEM) containing fetal bovine serum (10%) and penicillin/streptomycin (1%) at 37 °C for 24 h under 5% CO_2_. ZIF-8/PVA/PDA were added and further incubated at 37 °C under 5% CO_2_. Medium-containing materials were removed and the cell samples were treated with MTT for another 4 h after 24 h. To dissolve the formazan crystals, dimethyl sulfoxide (DMSO) was added. A Bio-Rad model-680 microplate reader was used to measure the absorbance at a wavelength of 490 nm with 630 nm as the reference wavelength. Results were obtained as the mean values by three measurements.

## 4. Conclusions

We used P-ZIF-8 synthesized in the water phase as an antibacterial material. P-ZIF-8 had a clear advantage over D-ZIF-8 synthesized in methanol in inhibiting foodborne Gram-positive *S. aureus* and it also effectively inhibited *E. coli*. Further analysis indicated that the antibacterial effect of P-ZIF-8 is caused by both ROS generation and dissociation of nanoparticles to release active Zn compounds. The spontaneous dissociation of P-ZIF-8 in aqueous solution enhances the inactivation of *S. aureus*, providing a theoretical basis for the development of further antibacterial materials. Additionally, we manufactured a ZIF-8-Film based on P-ZIF-8. Following a series of evaluations, ZIF-8-Film was found to have excellent performance and application potential in food safety and protection. ZIF-8-Film is a novel food packaging material that has strong intrinsic antibacterial activity and protective characteristics, which can spontaneously inactivate microorganisms on the surface while being reusable and exhibiting low toxicity. The excellent performance of ZIF-8-Film indicates that it may have potential as active packaging material for food preservation applications in the near future.

## Figures and Tables

**Figure 1 ijms-23-07510-f001:**
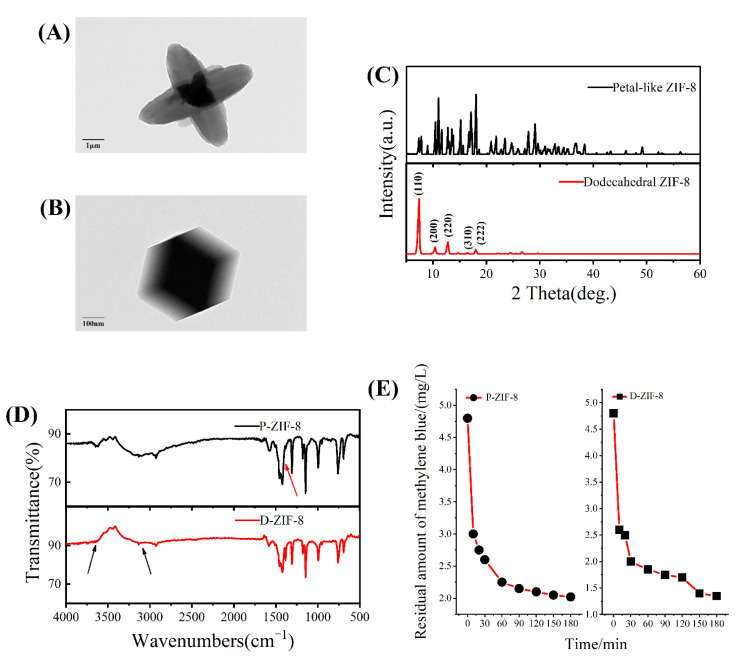
(**A**) TEM image of P-ZIF-8; (**B**) TEM image of D-ZIF-8; (**C**) The XRD diffraction pattern of P-ZIF-8 and D-ZIF-8; (**D**) The FTIR pattern of D-ZIF-8 and P-ZIF-8; (**E**) The degradation curve of P-ZIF-8 and D-ZIF-8 degrading methylene blue within 3 h.

**Figure 2 ijms-23-07510-f002:**
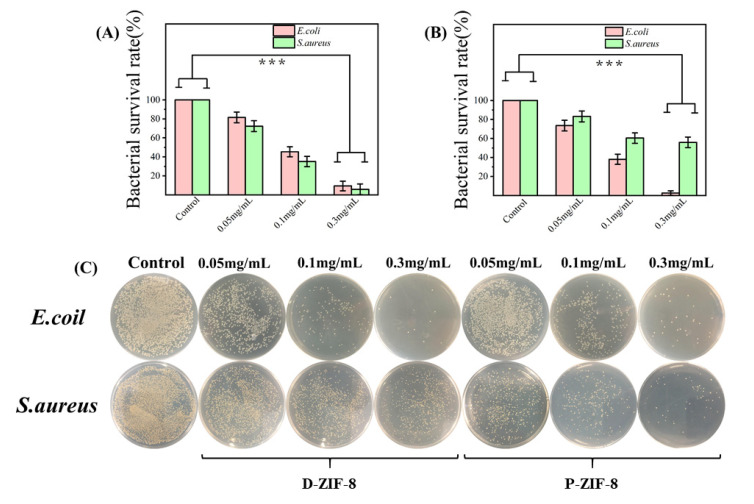
In terms of the number of surface atoms per milliliter, survival rates of bacteria treated with varying doses of ZIF-8. The real antibacterial properties of ZIF-8 were assessed using the colony-forming unit counting technique. The results show the averages and standard deviations from three separate trials. (**A**) Bacterial survival rate after P-ZIF-8 treatment; (**B**) Bacterial survival rate after D-ZIF-8 treatment (*** *p* < 0.001); (**C**) Bacterial colony images after treatment with the two forms of ZIF-8 mentioned above.

**Figure 3 ijms-23-07510-f003:**
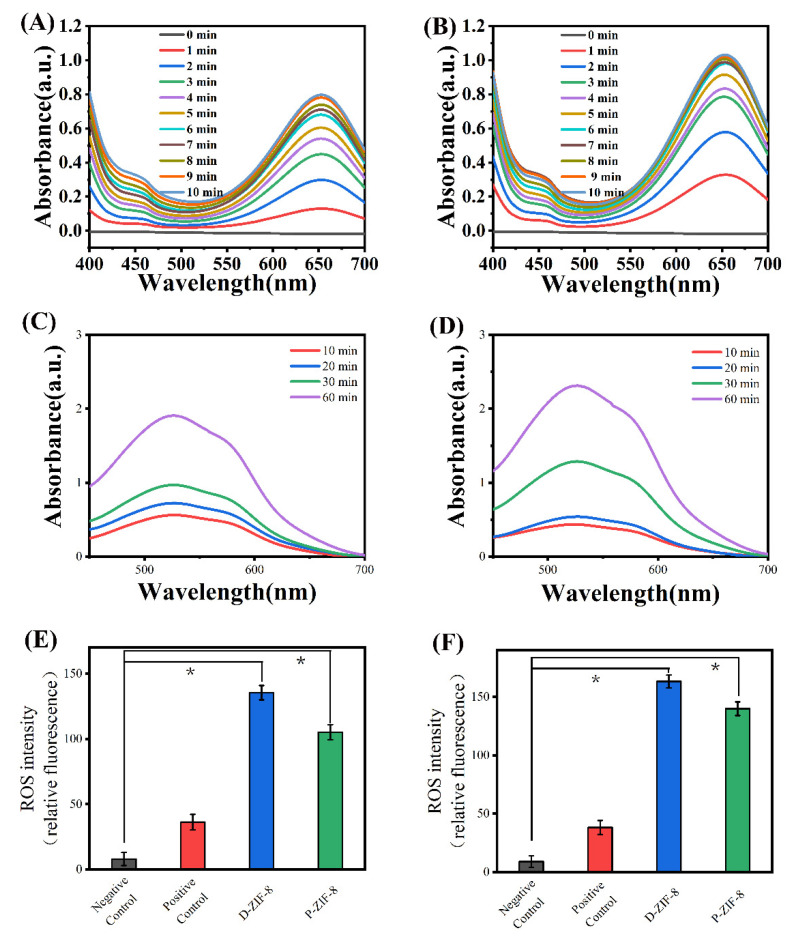
(**A**) Detection of the content of H_2_O_2_ produced by P-ZIF-8 under light; (**B**) Detection of the content of H_2_O_2_ produced by D-ZIF-8 under light; (**C**) Detection of the content of •O_2_^−^ produced by P-ZIF-8 under light; (**D**) Detection of the content of •O_2_^−^ produced by D-ZIF-8 under light; (**E**) Detection of intracellular ROS in *E. coli*; (**F**) Detection of intracellular ROS in *S. aureus.* (* *p* < 0.05).

**Figure 4 ijms-23-07510-f004:**
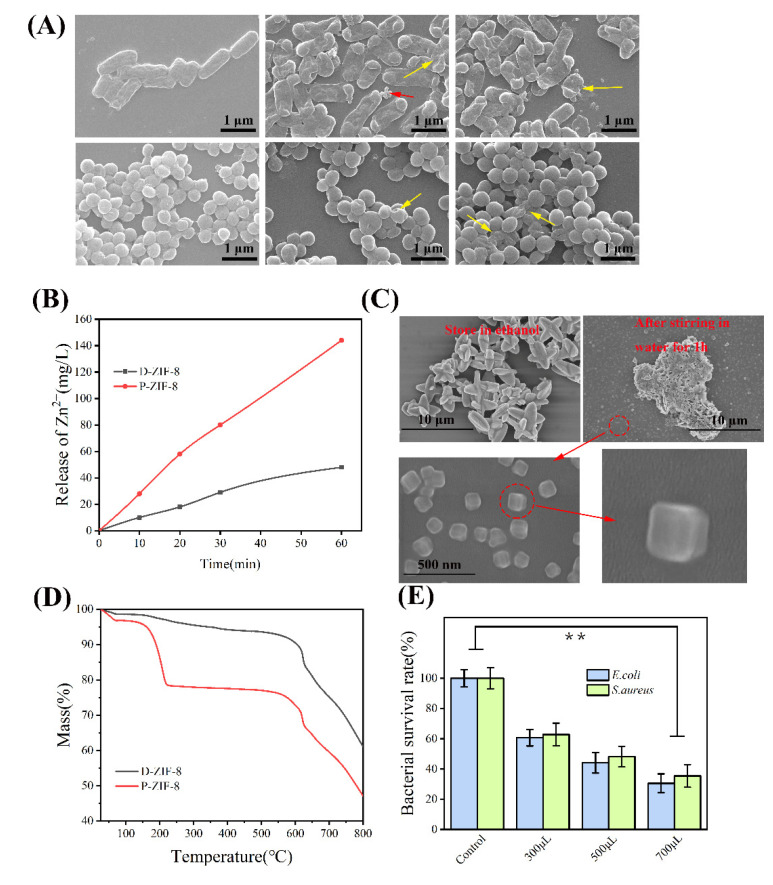
(**A**) Typical bacteria subjected to various treatments as seen using a scanning electron microscope. They were subjected to water, D-ZIF-8 and P-ZIF-8, in that order. *E. coli* and *S. aureus* are pictured from top to bottom. The injured bacteria are indicated by yellow arrows while surviving D-ZIF-8 is indicated by red arrows and possible remnants of P-ZIF-8 are indicated by yellow arrows. It is clear that it can still have its original form; (**B**) The ability of different forms of ZIF-8 to release Zn^2+^ in aqueous solution measured by ICP; (**C**) P-ZIF-8 stored in different solvents; (**D**) TG of different forms of ZIF-8; (**E**) Distinct antibacterial activity of Zn^2+^. Asterisks represent statistically significant differences (** *p* < 0.01).

**Figure 5 ijms-23-07510-f005:**
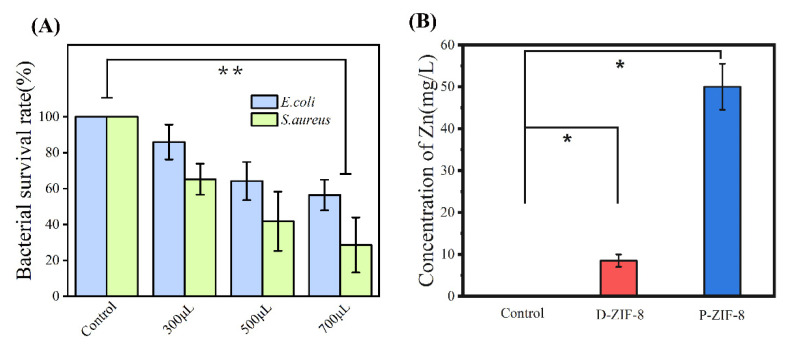
(**A**) Antibacterial ability of MOF extract; (**B**) The content of Zn detected in fragmented *S. aureus.* (* *p* < 0.05, *** p* < 0.01).

**Figure 6 ijms-23-07510-f006:**
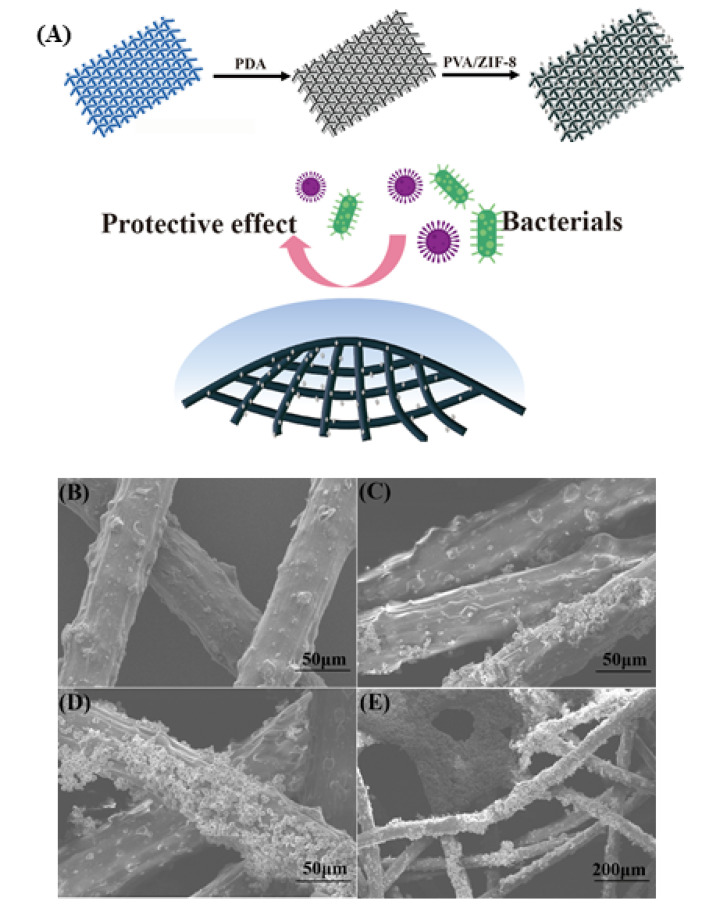
(**A**) Synthetic diagram of ZIF-8-Film; (**B**–**E**) The SEM image of ZIF-8-Film and fiber: (**B**) The SEM image of fiber; (**C**)The SEM image of fiber only with P-ZIF-8; (**D**,**E**) The SEM image of ZIF-8-Film.

**Figure 7 ijms-23-07510-f007:**
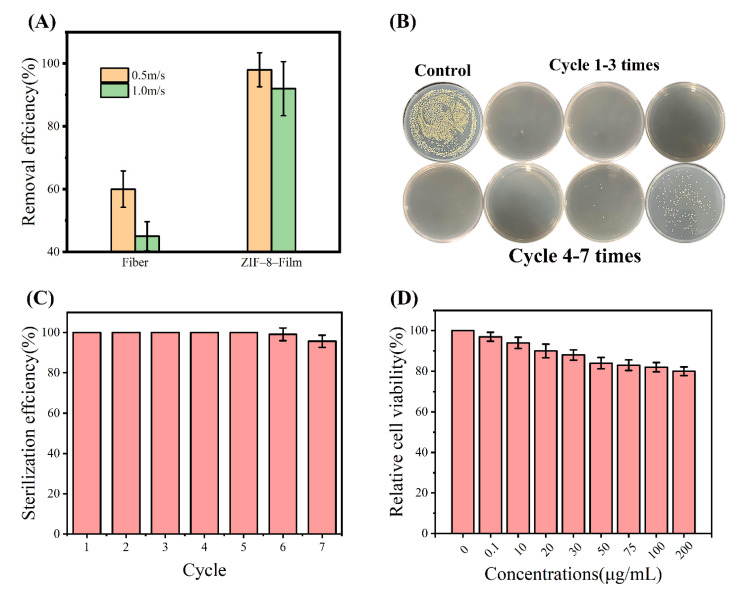
(**A**) Comparison of the particulate matter (PM) filtration efficiency between ZIF–8–Film and fiber; (**B**) Colony diagram of ZIF–8–Film repeated antibacterial test (**C**) Air disinfection performance of ZIF-8-Film continuously used for seven cycles. The error bars are calculated by repeating the measurements three times; (**D**) Cell viability of 3T3 cells treated with ZIF-8/PVA/PDA.

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
