# Peer review of "A Cruciform Petal-like (ZIF-8) with Bactericidal Activity against Foodborne Gram-Positive Bacteria for Antibacterial Food Packaging"

_ijms, 2022, doi:10.3390/ijms23147510_

Round 1
Reviewer 1 Report
Comments to the Author
The manuscript “A cruciform petal-like (ZIF-8) with bactericidal activity against foodborne Gram-positive bacteria for antibacterial food packaging” by Bowen Shen et al. would analyze the inhibitory effect on Gram positive bacteria of the cruciform petal-like zeolite imidazole framework-8 (ZIF-8).
I have several concern about the paper. In my opinion, although the relevance of the argument, the study requires substantial improvements both in the sentence construction and in the English grammar (please verify the style and typos).
Below are outlined the main points of revision.
Introduction:
The introduction provides insufficient background and lacks relevant references. The mechanism of action of the traditional ZIF-8 materials should also be explained.
Materials and Methods:
I suggest to move the supplementary materials in the whole text.
Discussion:
The discussion should be improved to highlight the importance of the obtained results and their application.
Author Response
Detailed Responses to Reviewers
Dear Editor/Reviewers,
Thank you very much for your kind suggestions, which will ensure our manuscript to be the best possible one. Those comments are all valuable and very helpful for revising and improving our paper, as well as the important guiding significance to our researches. We have made a corresponding revision of the manuscript. The manuscript has been totally revised and addressed as follows.
- please verify the style and typos
Response: Thanks for your advice. we have verified the typos and style in the manuscript and corrected it in the manuscript. Details are as follows:
futhrer→further(Line 31),
expecially→especially(Line 42),
Although→Althought(Line 49)
more easier→easier(Line 65),
illustrate→demostrates(Line 290),
20 μL drop on typical typical→20 μL drop on typical fiber (Line 391),
Long term→Long-term(Line 419,Line 424).
- The introduction provides insufficient background and lacks relevant references. The mechanism of action of the traditional ZIF-8 materials should also be explained.
Response: Thanks for your advice. First, the background and the relevant references have been added as you requested. (Line39-47)
“The World Health Organization showed that food safety has become a serious public health problem all over the world [1,2], and foodborne bacteria contaminating food was a common food safety issue [3-4]. It turns out that preventing foodborne bacteria had become even more difficult since the overuse of antibiotics and the subsequent rise of drug resistance especially represented by Gram-positive Staphylococcus aureus [5-6]. Therefore, nanomaterials have gradually become new antibacterial materials, and antibacterial films based on nanomaterials are gradually being used in food packaging [7]. Antibacterial food packaging systems can inhibit the growth of bacteria to preserve the safety and quality of the foods in the procedure of transportation and storage, which represents the innovative and promising food packaging technology [8, 9].”
- 1. Ali, A. N. M. A. Food safety and public health issues in Bangladesh: a regulatory concern Eur. Food Feed. Law Rev. 2013, 31-40.
- Fukuda, K. Food safety in a globalized world. Bull. World Health Organ. 2015, 93, 212-212.
- Narayanan, K.; Park, G.; Han, S. Biocompatible, antibacterial, polymeric hydrogels active against multidrug-resistant Staphylococcus aureus strains for food packaging applications. Food Control 2021, 123.
- Alvarez-Ordóñez, A.; Broussolle, V;. Colin, P;. Nguyen-The, C;. Prieto, M. The adaptive response of bacterial food-borne pathogens in the environment, host and food: implications for food safety. Int. J. Food Microbiol. 2015, 213, 99-109.
- Lynch, M.; Tauxe, R.; Hedberg, C. The growing burden of foodborne outbreaks due to contaminated fresh produce: risks and opportunities. Epidemiol Infect 2009, 137, (3), 307-15.
- Zhou, C.; Koshani, R.; OBrieen, B.; Ronholm, J.; Cao, X.; Wang, Y. Bio-inspired mechano-bactericidal nanostructures: a promising strategy for eliminating surface foodborne bacteria. Curr. Opin. Food Sci. 2021, 39, 110-119.
- Omerović, N.; Djisalov, M.; Živojević, K.; Mladenović, M.; Vunduk, J.; Milenković, I.; Knežević, N.; Gadjanski, I.; Vidić, J. Antimicrobial nanoparticles and biodegradable polymer composites for active food packaging applications. COMPR REV FOOD SCI F. 2021, 20, 2427-2454.
- Han, J. A review of food packaging technologies and innovations. Innovations in food packaging. 2014, 3-12.
- Omerović, N.; Djisalov, M.; Živojević, K.; Mladenović, M.; Vunduk, J.; Milenković, I.; Knežević, N. Ž.; Gadjanski, I.; Vidić, J. Antimicrobial nanoparticles and biodegradable polymer composites for active food packaging applications. COMPR REV FOOD SCI F. 2021, 20, (3), 2428-2454.
Then, we have added the relevant content about mechanism of action of the traditional ZIF-8 materials and some references in the manuscript as you requested. (Line55-60)
Traditional ZIF-8 has potent photo induced bactericidal activity, which can effectively kill Escherichia coli (E. coli) due to ROS generation in the presence of light [19]. Furthermore, traditional ZIF-8 can also release zinc ions and 2-methylimidazole to kill E. coli effectively under a variety of environmental conditions [20]. However, light-driven inactivation of bacteria as a photocatalyst is the main source of antibacterial ability for traditional ZIF-8 [21-23].
- Li, P.; Li, J.; Feng, X.; Li, J.; Hao, Y.; Zhang, J.; Wang, H.; Yin, A.; Zhou, J.; Ma, X. Metal-organic frameworks with photocatalytic bactericidal activity for integrated air cleaning. Nat. Commun. 2019, 10, (1), 1-10.
- Taheri, M.; Ashok, D.; Sen, T.; Enge, T. G.; Verma, N. K.; Tricoli, A.; Lowe, A.; Nisbet, D. R.; Tsuzuki, T. Stability of ZIF-8 nanopowders in bacterial culture media and its implication for antibacterial properties. Chem Eng J. 2021, 413, 127511.
- Pei, Zhen.; Fei, P.; Zhang, A.; Guo, J.; Hao, J.; Jia, J.; Dong, H.; Shen, Q.; Wei, L.; Jia, H.; Xu, B. Thermal oxygen sensitization modification and its visible light catalytic antibacterial performance for ZIF-8. J. Alloys Compd. 2022, 904, 164055.
- Liang, Z.; Wang, H.; Zhang, K.; Ma G.; Zhu, L.; Zhou, L.; Yan B. Oxygen-defective MnO2/ZIF-8 nanorods with enhanced antibacterial activity under solar light. Chem Eng J. 2022, 428, 131349.
- Kalati, M and Kamran A. Optimizing the metal ion release and antibacterial activity of ZnO@ ZIF-8 by modulating its synthesis method. New J. Chem. 2021, 45, 22924-22931.
- I suggest to move the supplementary materials in the whole text.
Response: According to your suggestion, we have moved the methods from the supplementary materials into the manuscript. (Line129-186)
- The discussion should be improved to highlight the importance of the obtained results and their application.
Response: We have rewritten the discussion section in order to connect the results to the application. (Line412-413, 438-439 and 443-447)
ZIF-8 Film has excellent filtration and waterproof ability, which is usually necessary for food packaging, and which means ZIF-8film has unique advantages in food packaging. (Line 412-413)
Usually, the antibacterial activity of food packaging can keep food fresh and safe from bacteria. (Line 438-439)
This means that our ZIF-8 Film is also safe for the human body, and there is no need to consider the harm to the human body caused by the residue of food packaging. All in all, our ZIF-8 Film can remove harmful bacteria, maintain the freshness and safety of food, and more importantly, ZIF-8 Film shows the characteristics of being harmless to the human body, eliminating people's concerns about the safety of food packaging. (Line 443-447)

Reviewer 2 Report
The paper is focused on the development of antibacterial nanomaterials suitable for food packaging applications. The topic falls within the scope of the journal. I recommend the publication after the following revisions:
- Experimental details for TGA measurements should be reported. As examples, gas flows for both balance and sample, heating rate and temperature interval should be indicated in the paragraph 2.3.
- The scale length in SEM images presented in Fig. 4A is not well visible. Please check and revise.
- Figure 4D. As concerns TG curves, y-axis should be indicated as “Mass / %” insetad of “Weight loss / %” being that the curve starts from 100.
- The presentation and discussion of TG data should be improved. The authors could estimate the mass losses from 25 to 150 °C (moisture loss). Moreover, the residual masses at 800 °C could be estimated. I suggest to present the corresponding differential thermogravimetric (DTG) curves. I recommend to compare the temperatures at DTG peaks for the investigated sample to evidence their different thermal stability.
- Introduction could be updated by evidencing that antibacterial composite materials can be obtained by incorporation of silver nanoparticles [DOI: 10.1016/j.colsurfa.2022.128525] and essential oils [DOI: 10.1016/j.carbpol.2016.07.041] within inorganic nanoclays that can be filled within polymeric films.
Author Response
Detailed Responses to Reviewers
Dear Editor/Reviewers,
Thank you very much for your kind suggestions, which will ensure our manuscript to be the best possible one. Those comments are all valuable and very helpful for revising and improving our paper, as well as the important guiding significance to our researches. We have made a corresponding revision of the manuscript. The manuscript has been totally revised and addressed as follows.
- Experimental details for TGA measurements should be reported. As examples, gas flows for both balance and sample, heating rate and temperature interval should be indicated in the paragraph 2.3.
Response: According to your suggestion, we have added the experimental details of TGA in the paragraph 2.3. (Line 106-107)
TGA was performed in air with an alumina crucible on a Netzsch STA 449 F3 thermogravimetric analyzer, among them, the mass of the sample was 30 mg, the heating rate was 10 ℃/min. (Line 106-107)
- The scale length in SEM images presented in Fig. 4A is not well visible. Please check and revise.
Response: We have added scale length to the SEM images in Fig. 4A. The updated image has been uploaded to the manuscript.
- Figure 4D. As concerns TG curves, y-axis should be indicated as “Mass / %” insetad of “Weight loss / %” being that the curve starts from 100.
Response: We have corrected the Y-axis error at TG curves as asked. The updated image has also been uploaded to the manuscript.
- The presentation and discussion of TG data should be improved. The authors could estimate the mass losses from 25 to 150 °C (moisture loss). Moreover, the residual masses at 800 °C could be estimated. I suggest to present the corresponding differential thermogravimetric (DTG) curves. I recommend to compare the temperatures at DTG peaks for the investigated sample to evidence their different thermal stability
Response: The description of TG has been added in the manuscript according to the suggestion. (Line 338-340)
D-ZIF-8 lost about 2% of water at 25-150°C, while P-ZIF-8 lost about 15% of crystal water at 25-150°C. The result at a high temperature of 800°C was that D-ZIF-8 had 60% of its weight remaining, while P-ZIF-8 had less than 50% remaining. (Line 338-340)
The picture of DTG has been added in the Supplementary Materials (Fig S5), and the description of DTG has been added in the manuscript. (Line340-342)
The DTG results showed that P-ZIF-8 had two distinct thermal decomposition activities, while D-ZIF-8 had only one, which indicated that P-ZIF-8 had worse thermal stability (Fig. S5). (Line 342-344)
- Introduction could be updated by evidencing that antibacterial composite materials can be obtained by incorporation of silver nanoparticles [DOI: 10.1016/j.colsurfa.2022.128525] and essential oils [DOI: 10.1016/j.carbpol.2016.07.041] within inorganic nanoclays that can be filled within polymeric films.
Response: We have added the antibacterial ability of the complex to the text, as well as the relevant literatures. (Line 420-423) (Reference 52-53)
The ZIF-8-Film should not only be able to filter germs, but it should also be highly antimicrobial. As a result, we focused on the antibacterial properties and long long-term use performance of the ZIF-8-Film. As previously reported, the combination of nanomaterials and polymers tends to have better antibacterial capabilities [52, 53]. (Line 420-423)
- Shevtsova, T.; Caxallaro, G.; Lazzara, G.; Milioto, S.; Donchak, V.; Harhay, K.; Korolko, S.; Budkowski, A.; Stetsyshyn, Y. Temperature-responsive hybrid nanomaterials based on modified halloysite nanotubes uploaded with silver nanoparticles. Colloid Surface A. 2022, 641, 128525.
- Biddeci, G.; Cavallaro, G.; Di Blasi, F.; Massaro, M.; Milioto, S.; Parisi, F.; Riela, S.; Spinell, G. Halloysite nanotubes loaded with peppermint essential oil as filler for functional biopolymer film. Carbohydr. Polym. 2016, 152, 548-557.

Round 2
Reviewer 1 Report
The authors have addressed the required issues. Just pay attention to the zif8-film name: sometime is written Zif8-film and sometime Zif8-Film.
Reviewer 2 Report
The paper was correctly revised as suggested by the reviewers. I recommend its publication in the present form.